# Safety of Early Bevacizumab Administration after Central Venous Port Placement for Patients with Colorectal Cancer

**DOI:** 10.3390/cancers15082264

**Published:** 2023-04-12

**Authors:** Hirona Shigyo, Hiroyuki Suzuki, Toshimitsu Tanaka, Etsuko Moriyama, Yasutaka Shimotsuura, Sachiko Nagasu, Hideki Iwamoto, Yoshito Akagi, Kenta Murotani, Takumi Kawaguchi, Keisuke Miwa

**Affiliations:** 1Multidisciplinary Treatment Cancer Center, Kurume University Hospital, Kurume 830-0011, Japan; 2Department of Surgery, Kurume University School of Medicine, Kurume 830-0011, Japan; 3Division of Gastroenterology, Department of Medicine, Kurume University School of Medicine, Kurume 830-0011, Japan; 4Biostatistics Center, Kurume University, Kurume 830-0011, Japan

**Keywords:** central venous port, bevacizumab, complication, wound healing, colorectal cancer

## Abstract

**Simple Summary:**

The safety of bevacizumab administration immediately after central venous (CV) port placement, a minor surgery, is still unknown. We investigated whether Bevacizumab is safe when administered early after CV port placement. Bevacizumab therapy after CV port implantation did not affect the frequency of complications, regardless of the timing of initiation. These results support that bevacizumab is safe when administered early after CV port placement.

**Abstract:**

Bevacizumab (BEV) requires an adequate withdrawal period to avoid BEV-related complications during major surgery. However, the safety of BEV administration immediately after surgical placement of the central venous (CV) port, a minor surgery, is still unclear. This study aimed to investigate whether BEV is safe when administered early after CV port placement. We retrospectively evaluated 184 patients with advanced colorectal cancer (CRC) treated with a BEV-containing regimen and divided them into two groups according to the interval between CV port implantation and chemotherapy initiation, with the early administration group being ≤7 days and late administration group being >7 days. Complications were then compared between the two groups. The early-administration group was significantly older and had a higher rate of colon cancer than the late-administration group. Overall, 24 (13%) patients developed CV port-related complications. Male sex was a risk factor for complications (odds ratio [OR], 3.154; 95% CI, 1.19–8.36). The two groups showed no significant difference in the frequency of complications (*p* = 0.84) or patient characteristics (after the inverse probability of treatment weighting, *p* = 0.537). In conclusion, the frequency of complications is not affected by the timing of BEV initiation after CV port implantation. Thus, early BEV administration after CV port placement is safe.

## 1. Introduction

Colorectal cancer (CRC) is one of the most common digestive system malignancies [1]. Although advances in treatment strategies for unresectable CRC, such as chemotherapy, targeted therapy, and immunotherapy, have improved these patients’ prognoses, more than 800,000 deaths are still reported annually [2]. Bevacizumab (BEV), a monoclonal antibody against vascular endothelial growth factor (VEGF) that inhibits VEGF signaling, was first approved for patients with metastatic CRC in 2004 [3] and then extensively used in broad types of cancer treatments [4]. By blocking the biological activity of VEGF, BEV suppresses tumor growth via inhibition of angiogenesis in the tumor tissues and also affects immune cells such as macrophages and lymphocytes [4]. Nowadays, BEV is one of the most important drugs in standard therapy for CRC [5,6]. The placement of long-term venous access devices (e.g., subcutaneously implanted central venous (CV) access devices), first introduced in 1982, is a common requirement in the management of oncologic patients and is widely used, especially for establishing intravenous access for chemotherapy [7]. CV port implantation allows for reliable drug delivery and reduces patient discomfort from route complications during chemotherapy administration [8]. In addition, some chemotherapy regimens for CRC require 48 h of continuous intravenous infusion, and CV port placement is mandatory in such cases [9]. BEV inhibits VEGF signaling, and thus, chemotherapy regimens using BEV should be withdrawn approximately 4 weeks before a major surgical procedure to prevent BEV-related complications, such as protracted wound healing and gastrointestinal (GI) perforation/hemorrhage [10]. Although the placement of a CV port is not generally considered major surgery, there are few reports on the safety of starting a BEV-containing regimen immediately after CV port implantation. Within 10–14 days after CV port implantation, the frequency of wound dehiscence may increase, as reported previously [11,12]. Meanwhile, a study that evaluated 20 patients in whom a CV port was implanted during BEV therapy has shown that there are no adverse events despite the early administration of BEV [13]. Japanese guidelines for CV port placement have not reached a consensus on when is the appropriate BEV administration time after CV port implantation [14]. In some opinions about CV port placement, it could be performed while patients are receiving BEV, while other opinions hold that bevacizumab should be withdrawn for 7 days before and after the procedure. The safety of starting a BEV-containing regimen immediately after CV port implantation is still unclear and controversial. Thus, this study aimed to verify that BEV-containing regimens are safe when administered early after CV port placement.

## 2. Materials and Methods

### 2.1. Study Design and Patients

This retrospective study evaluated 241 patients who were pathologically diagnosed with unresectable CRC and received CV port implantation for reliable drug delivery for chemotherapy between April 2014 and March 2022 at Kurume University Hospital, Kurume, Japan. The inclusion criteria were as follows: (1) histologically diagnosed advanced CRC; (2) age ≥ 18 years; (3) Eastern Cooperative Oncology Group performance status (ECOG-PS) score ≤ 2; and (4) life expectancy ≥ 12 weeks. The exclusion criteria were as follows: (1) brain metastases or spinal cord compression; (2) poor general condition (ECOG-PS score ≥ 3); (3) poorly controlled infection; and (4) previous GI hemorrhage or ileus in the month before the beginning of the study and/or high risk of bleeding or ileus. We divided the patients into two groups according to the interval between CV port implantation and the start of chemotherapy: early administration group, ≤7 days and late administration group, >7 days. CV port system-related and BEV-related complications were then compared between the two groups.

### 2.2. Data Collection

The following data were collected from medical charts and reviewed: age; sex; body mass index (BMI); ECOG-PS score; anticoagulant medication; type of comorbid malignancies; cancer stage according to the 8th edition of the Union for International Cancer Control (UICC) TNM classification; serum C-reactive protein level, carcinoembryonic antigen level; carbohydrate antigen 19-9 level; and comorbidities and complications including diabetes mellitus (defined as fasting plasma glucose ≥126 mg/dL and hemoglobin A1c ≥ 6.5% [15]), hypertension (defined as a systolic blood pressure ≥140 mmHg and/or a diastolic blood pressure ≥90 mmHg [16]), hyperlipidemia (defined as serum triglyceride levels of ≥240 mg/dL and/or serum low-density lipoprotein cholesterol levels of ≥160 mg/dL) [17]), and chronic kidney disease (CKD; defined as CKD stage ≥ 3).

### 2.3. CV Port Placement

CV ports were implanted under fluoroscopic and ultrasound guidance, with maximum sterile precautions in all patients. The CV port kit used in this study was (1) a 6-Fr. ChronoFlex™ catheter and PowerPort MRI (Becton, Dickinson and Company, Franklin Lakes, NJ, USA); (2) a 6-Fr. ChronoFlex™ catheter and PowerPort^®^ ClearVUE^®^ isp (Becton, Dickinson and Company); or (3) a Safe Guide™ Micro Needle Port (Cardinal Health, Dublin, OH, USA). The catheter was inserted into the right subclavian or internal jugular vein on the left or right side and then passed under the skin. The port was implanted in a subcutaneous pocket of the chest wall on the same side as the venipuncture. Regarding the CV port placement procedure, information on the position of the catheter tip, time taken for the procedure, and amount of blood loss was collected.

### 2.4. Complications and Event-Free Survival

CV port complications were defined as adverse events that occurred during or after CV port implantation. Complications were divided into CV-port-related complications (i.e., infection, catheter damage, catheter migration, catheter embolization, and port site skin damage) and BEV-related complications (i.e., subcutaneous hematoma, erosions or ulcers, wound dehiscence, delayed wound healing, GI perforation, and fistula). Early complications were defined as those that occurred within 30 days of implantation. Event-free survival was defined as the time in months between CV port implantation and the event/last follow-up date. An event was defined as a CV port complication if it occurred during the follow-up period.

### 2.5. Statistical Analyses

Data are presented as the median (range) or n (%). Each parameter was compared between groups of early administration and late administration using Fisher’s exact test for categorical variables and the Mann–Whitney U-test for continuous variables. The propensity score was calculated using logistic regression according to 8 factors as follows: age, procedure time, ECOG-PS score, diabetes mellitus, hypertension, hyperlipidemia, CKD, and anticoagulant medication. Adjustment by propensity score was conducted using inverse probability of treatment weighting (IPTW) and a matching method according to three ratios: 1:1. The influencing factors of the incidence of complications were evaluated using univariate analysis. Kaplan–Meier curves were plotted to determine the event-free survival rate in patients with advanced CRC for whom early/late administration of BEV-containing chemotherapy was started who were treated with chemotherapy regimens with/without BEV. Stratified log-rank *p* values (two-sided) were calculated for time-to-event endpoints. All statistical analyses were performed using JMP Pro version 15.0 software and SAS9.4 (SAS Institute, Inc., NC, USA). A *p* less than 0.05 was considered statistically significant.

## 3. Results

### 3.1. Patient Characteristics

Among the 241 patients with advanced CRC who were initially evaluated, we classified 57 patients as historical controls because their chemotherapy regimen did not include BEV. Ultimately, 184 patients were analyzed who were treated with BEV-containing regimens in the study. The patient selection flowchart is shown in Figure 1. The groups early BEV administration and late BEV administration involved 88 and 96 patients, respectively. The baseline clinicodemographic characteristics are listed in Table 1. The median age was 67 years (range, 32–85 years), and 52% (96/184) of the patients were female. The majority of patients in the study had an ECOG-PS score of ≤1, whereas only 6% (12/184) of the patients had an ECOG-PS score of 2. Patients in the early administration group were significantly older (68 years vs. 66 years, *p* = 0.018) and had a significantly higher rate of colon cancer than those in the late administration group (72% vs. 56%, *p* = 0.03). There were no significant differences in the other parameters such as sex; ECOG-PS score; anticoagulant medication use; BMI; or comorbidities including diabetes mellitus, hypertension, hyperlipidemia, and CKD between the groups.

### 3.2. CV Port Characteristics and Outcomes

The technical success rate for CV port placement was 100%. Of the 184 insertions, 169 (92%) were performed via the right internal jugular vein, 11 (6%) via the left internal jugular vein, and 4 (2%) via the subclavian vein (Table 2). The median procedure time was 40 min (range, 16–65 min), and the median catheter indwelling period was 521 days (range, 2–3014 days). PowerPort MRI was used in 129 patients (70%), PowerPort^®^ ClearVUE^®^ isp in 32 patients (17%), and Safe Guide™ Micro Needle Port in 23 patients (13%) (Table 2). There were no significant differences in the insertion vein, procedure time, or device between the groups of early administration and late administration. The catheter indwelling period in the late administration group was significantly longer than that in the early administration group [441.5 (2–2163) minutes vs. 641 (5–3014) minutes, *p* < 0.01] (Table 2).

### 3.3. Complications during or after CV Port Implantation

Overall, 13% of the patients (24/184) developed CV port-related complications (Figure 2A). There were no significant differences in total complication occurrence rates between the early administration (13%, 11/88) and the late administration groups (14%, 13/96) (*p* = 0.84) (Figure 2A). There were two (1.1%) patients who developed early periprocedural complications. The median time to complication onset was 367.5 days (range, 2–1467 days) (Table 3). The complications included catheter/port system-related complications (n = 24), infection (n = 6), catheter damage (n = 5), catheter migration (n = 1), catheter embolization (n = 1), and port site skin complications (n = 10) (Table 3). There were no BEV-related complications, including subcutaneous hematoma, erosions or ulcers, wound dehiscence, delayed wound healing, GI perforation, or fistula. There were no significant differences in the total complications, early complications, and median time to complication onset between the groups of early administration and late administration (Table 3). In 22 patients (12%), CV port-related complications made it difficult to continue therapy, and the CV port was thus removed. Two patients with catheter embolization were continued on anticoagulant medications. Kaplan–Meier curve analysis for event-free survival revealed no significant differences in event-free survival rates between the early administration and the late administration groups (log-rank *p* = 0.511) (Figure 2B).

### 3.4. Propensity Score Matching Analysis and Risk Factors of Complications

Table 4 shows the patient characteristics before adjustment (n = 184, odds ratio (OR), 0.912; 95% confidence interval (CI), 0.386–2.157), after adjustment (n = 138, OR, 1.185; 95% CI, 0.377–3.727), and after IPTW (n = 184, OR, 0.822; 95% CI, 0.442–1.530). No significant differences were observed in any patient characteristics. Univariate analysis revealed that the male sex was a significant risk factor for developing CV port implantation complications during or after (OR, 3.154; 95% CI, 1.19–8.357, *p* = 0.021) (Appendix A).

### 3.5. Comparision of Complication Rates with/without BEV-Containing Chemotherapy Regimen

To assess whether the complication rate could be elevated by chemotherapy with or without BEV, we evaluated the patients who did not receive a BEV-containing chemotherapy regimen in the same period as the historical controls (Figure 1). No significant differences were observed in the baseline characteristics (Appendix A). Then, we performed the Kaplan–Meier curve analysis for event-free survival in patients with CRC who were treated with a BEV-containing chemotherapy regimen (n = 184) or not (n = 57) (Appendix A). There were no significant differences in event-free survival rates between these two groups (log-rank *p* = 0.889).

## 4. Discussion

The safety of starting a BEV-containing regimen administered immediately after CV port implantation has not been established to date. This study found that the initiation period of BEV therapy after CV port implantation did not affect the frequency of complications. Furthermore, propensity score matching analysis showed that the timing of BEV therapy initiation did not affect the occurrence of complications.

Folinic acid, 5-fluorouracil (5-FU), oxaliplatin or folinic acid, 5-FU, and irinotecan are widely used in CRC for their antitumor efficacy and safety [18,19,20,21]. These regimens require CV port placement for the continuous intravenous administration of 5-FU at home [9]. In general, the longer the indwelling period, the higher the frequency of complications, such as bleeding and hematoma, wound dehiscence, infection, thrombosis, and catheter damage. In our study, complications associated with CV port implantation occurred in 13% of patients, and early complications (within 30 days) occurred in 1.1% of patients. Previous studies have reported an overall complication rate of 6.2–18.4% [22,23,24,25,26] and an early complication rate of 7–11.6% [27] in patients with CV port implantation. The overall incidence of complications in our study corresponds to that in previous studies, but the early complication rate was lower.

BEV is a recombinant IgG1 humanized monoclonal antibody against VEGF that specifically inhibits VEGF binding to its receptors [4]. By blocking the biological activity of VEGF, BEV suppresses angiogenesis in tumor tissues and inhibits its growth, and it also reduces vascular permeability and interstitial pressure in tumor tissue [4,28]. A chemotherapy regimen that contains BEV could be safely used even in patients with advanced CRC under hemodialysis [29]; however, it should be noted that wound dehiscence with BEV treatment was observed in a phase III trial of the treatment in patients with advanced CRC [3]. Therefore, an interval of 6–8 weeks between the last administration of BEV and surgery is recommended, and there should be at least a 4-week wait before BEV is initiated postoperatively. Minor surgeries, such as CV port implantation, have been discussed several times; however, there is no consensus regarding the recommended duration of drug withdrawal. Zawacki et al. reported that administration of BEV within 10 days of implantation resulted in a high rate of wound dehiscence [11]. Erinjeri et al. showed that the frequency of wound dehiscence was significantly higher in patients who used BEV within 14 days of implantation [12]. Meanwhile, Grenader, et al. evaluated 20 patients in whom a CV port was implanted during BEV therapy and found no adverse events despite the early administration of BEV [13]. Consistent with this report, there were no BEV-related complications regardless of the timing of BEV treatment initiation; additionally, we found that the frequency of total complications was not affected by BEV treatment initiation early after CV port implantation. In addition, propensity score matching analysis showed that the timing of BEV therapy initiation did not affect the occurrence of complications. Thus, early BEV administration after CV port placement might be safe. Collectively, these findings support that BEV does not affect wound healing even if it is administered early after CV port placement. Although this study has small cases with selection bias, there was no difference in complications evaluated with chemotherapy with or without BEV. Therefore, it is suggested that the administration of BEV itself might not affect whether CV port-related complications occur.

One reason for the lack of complications regardless of BEV therapy is procedural standardization and device improvements. Since the CV port was first introduced in 1982, various investigational trials, including on the materials or coatings of the catheter, choice of blood vessels, techniques, and use of intraoperative ultrasound and position of the catheter tip, have been conducted [30]. CV port devices have also been improved to facilitate placement, such as improving the shape of the puncture needle and the visibility of the catheter [30,31]. Furthermore, from the perspective of preventing infections, the catheter and port body have long-lasting antibacterial effects, while the Groshong catheter prevents thrombus formation [30,32]. As described in the guidelines [33], widespread knowledge of vessels that should not be used and the recommended approach methods have led to the standardization of safe and accurate techniques with fewer complications. With the widespread use of CV port procedures, these techniques have been standardized, simplified, and refined. Consistent with the fact that a shorter procedure time during CV port placement is associated with a lower incidence of complications [34], the median CV port procedure time of 40 min in our study might have been one of the reasons for the low number of complications. Consistent with a previous cohort study for central venous port failure in 1280 cancer patients that shows that being male is a risk factor for complications (HR, 1.566; 95%CI, 1.318–1.861) [35], being male was identified as a risk factor for occurrence of CV port complications in the current study. Possible explanations for this might include gender differences in the muscle mass of the port implantation site, the amount of daily activity, and the ability to keep the implantation site clean; however, further research is needed to accumulate on these factors.

Molecular targeted agents other than BEV, ramucirumab [36] and aflibercept [37], are the standard chemotherapeutic agents for patients with unresectable CRC. Considering that these agents commonly inhibit VEGF signaling, studies on CV port implantation and the frequency of complications are warranted for these agents as well. On the other hand, since these non-BEV agents are often used as second-line chemotherapy, and CV ports are often implanted at the time of first-line chemotherapy, therefore, drug administration immediately after CV port implantation might not be a major problem with these drugs.

The present study has the limitation of being a single-center, retrospective study. In addition, selection bias may exist because it is possible that the physician avoided administering the first dose of BEV owing to the risks involved. In addition, this study did not include patients who received BEV prior to CV port implantation. In the future, we will conduct a multicenter prospective study and conduct further investigations to include cases in which BEV was used before CV port implantation.

## 5. Conclusions

The frequency of complications might not be affected by the initiation of BEV therapy after CV port implantation regardless of the timing of initiation. These findings support that BEV is safe to administer early after CV port placement.

## Figures and Tables

**Figure 1 cancers-15-02264-f001:**
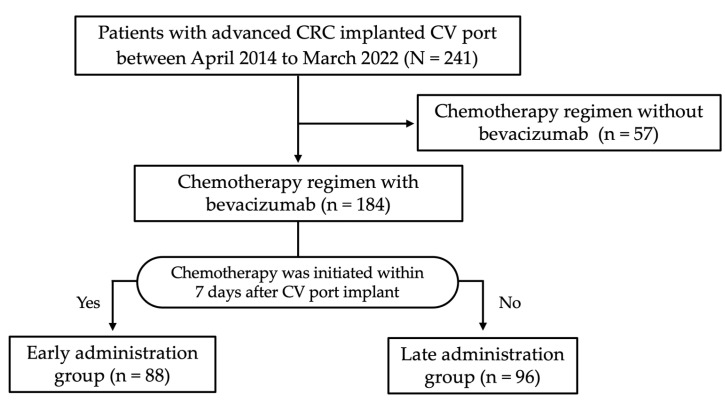
Consort flow chart diagram. Flowchart showing patient selection. Abbreviations: CRC: colorectal cancer; CV: central venous.

**Figure 2 cancers-15-02264-f002:**
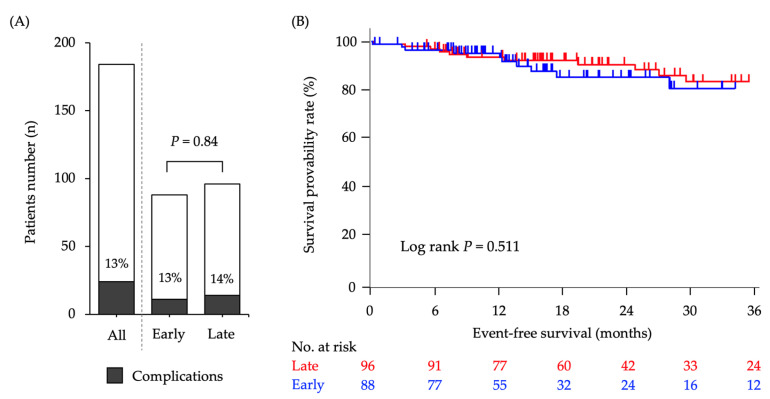
Complications during or after CV port implantation. (**A**) In total, 13% (24/184) of the patients developed CV port-related complications. The rate of complications was 13% (11/88) in the early administration group (early) and 14% (13/96) in the late group (late), with no significant difference between the two groups (*p* = 0.84). (**B**) Kaplan–Meier analysis for the early administration group (early) vs. the late administration group (late), with no significant difference between the two (*p* = 0.511).

**Table 1 cancers-15-02264-t001:** Baseline patient characteristics.

Characteristics	Chemotherapy Regimen with BEV (n = 184)	Early Administration Group (n = 88)	Late Administration Group (n = 96)	*p*
Age, years	67 [32–85]	68 [34–85]	66 [32–84]	**0.018**
Sex; female	96 (52)	46 (52)	42 (44)	0.25
Male	88 (48)	42 (48)	54 (56)	
Eastern Oncology Group Performance Status score				0.77
0	106 (58)	53 (60)	53 (55)	
1	66 (36)	30 (34)	35 (38)	
2	12 (6)	5 (6)	7 (7)	
Tumor site				**0.03**
Colon	117 (64)	63 (72)	54 (56)	
Rectum	67 (34)	25 (28)	42 (44)	
Hypertension				0.22
Yes	39 (21)	22 (25)	17 (18)	
No	145 (79)	66 (75)	79 (82)	
Hyperlipidemia				0.20
Yes	14 (8)	9 (10)	5 (5)	
No	170 (93)	79 (90)	91 (95)	
Chronic kidney disease				0.37
Yes	9 (5)	3 (3)	6 (6)	
No	175 (95)	85 (97)	90 (94)	
Diabetes				0.71
Yes	19 (10)	10 (11)	9 (10)	
No	165 (90)	78 (89)	87 (90)	
Anticoagulant medication				0.66
Yes	29 (16)	10 (11)	19 (20)	
No	155 (84)	78 (89)	77 (80)	
Body mass index	21.5 [16–42]	21.5 [16–29]	21.4 [16–42]	0.72

Data are presented as the n (%) or as the median [range]. Abbreviations: BEV: bevacizumab.

**Table 2 cancers-15-02264-t002:** Characteristics of CV port placement.

Characteristics	All (n = 184)	Early Administration Group (n = 88)	Late Administration Group (n = 96)	*p*
Insertion vein Right internal jugular Left internal jugular Right subclavian	169 (92) 11 (6) 4 (2)	83 (94) 5 (6) 0 (0)	86 (90) 6 (6) 4 (4)	**0.04**
Procedure time (minutes)	40 [16–65]	39.5 [16–60]	40 [21–65]	0.10
Catheter indwelling period (days)	541 [2–3014]	441.5 [2–2163]	641 [5–3014]	**<0.01**
Device type PowerPort^®^ MRI PowerPort^®^ ClearVUE^®^ isp Safe Guide™ Micro Needle Port	129 (70) 32 (17) 23 (13)	67 (76) 15 (17) 6 (7)	62 (64) 17 (18) 17 (18)	0.66

Data are presented as the n (%) or as the median [range].

**Table 3 cancers-15-02264-t003:** Complications during or after CV port implantation.

Subjects	All (n = 184)	Early Administration Group (n = 88)	Late Administration Group (n = 96)	*p*
Early complications (within 30 days) *	2 (1.1)	1 (1.1)	1 (1.0)	1.00
Time to complication onset (days)	367.5 [2–1467]	364 [2–1467]	371 [3–1349]	0.62
Catheter/port system-related complications				
Infection	6 (3.3)	3 (3.4)	3 (3.1)	1.00
Catheter damage	5 (2.7)	0 (0)	5 (5.2)	**0.06**
Catheter migration	1 (0.5)	1 (1.1)	0 (0)	0.48
Catheter embolization	2 (1.1)	0 (0)	2 (2.1)	0.50
Port site skin damage	10 (5.4)	7 (8.0)	3 (3.1)	0.20
Bevacizumab-related complications				
Subcutaneous hematoma	0	0	0	1.00
Erosions or ulcers	0	0	0	1.00
Wound dehiscence	0	0	0	1.00
Delayed wound healing	0	0	0	1.00
GI perforation	0	0	0	1.00
Fistula	0	0	0	1.00

Abbreviations: GI: gastrointestinal. *: One case of catheter migration and one case of catheter damage.

**Table 4 cancers-15-02264-t004:** Propensity score matching analysis.

Method	n	Odds Ratio	95% CI	*p*
Unadjusted	184	0.912	0.386–2.157	0.834
Matching	138	1.185	0.377–3.727	0.771
IPTW	184	0.822	0.442–1.530	0.537

Abbreviations: IPTW: inverse probability of treatment weighted; CI: confidence interval.

## Data Availability

The data presented in this study are available on request from the corresponding author.

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
