# Peer review of "Safety of Early Bevacizumab Administration after Central Venous Port Placement for Patients with Colorectal Cancer"

_cancers, 2023, doi:10.3390/cancers15082264_

Round 1
Reviewer 1 Report
Hirona Shigyo et al. conducted a study to address the clinically relevant question of whether bevacizumab can be administered to patients immediately after CV port insertion, which is a common scenario encountered in clinics treating colorectal cancer patients. This retrospective study evaluated 241 patients with unresectable colorectal cancer who received chemotherapy with BEV for the first time after CV port placement between April 2014 and March 2022 at Kurume University Hospital, Japan. The study aimed to compare CV port system-related and BEV-related complications between two groups of patients: those who received early administration of BEV (within seven days after CV port placement) and those who received late administration of BEV (more than seven days after CV port placement). The article details the study design, patient selection criteria, data collection methods, CV port placement procedures, and the types of complications observed. Statistical methods were used to analyze the data, and the results, including patient characteristics and the incidence of complications in the two groups, are presented.
Major comments:
1. There are no contraindications or cautions mentioned in the summary of product characteristics regarding minor surgery and treatment with bevacizumab.
2. The rationale for dividing the two groups based on more or less than seven days between minor surgery and administration of bevacizumab should be explained. The summary of product characteristics mentions 28 days for major surgery.
3. Consider using time-dependent event analysis by constructing Kaplan-Meier curves for patients receiving regimens with and without bevacizumab and comparing them using logrank test. Instead of dichotomizing +/- 7 days, consider using time data as a continuous variable.
4. The study design being retrospective may limit the strength of the conclusions drawn.
Minor comments:
1. The statement "BEV has shown antitumor effects in CRC and is one of the most important drugs in standard therapy for CRC" on line 47-48 requires justification by referencing several phase III study data or modification.
2. Please verify the number and fraction of females on line 127.
Author Response
We thank the reviewer for evaluating our manuscript. The following text describes our responses to the comments.
Response to the comments of Reviewer 1
Major comments:
- There are no contraindications or cautions mentioned in the summary of product characteristics regarding minor surgery and treatment with bevacizumab.
Response: Thank you for this insightful comment. As the reviewer pointed out, there are no contraindications or precautions for minor surgery and treatment with bevacizumab listed in the summary of product characteristics. Although it has been reported that early initiation of bevacizumab administration after CV port implantation increased the delay in wound healing, guidelines for CV port implantation in Japan have not reached a consensus [1]. In a guideline for the proper use of bevacizumab in Japan recommend a one-week drug withdrawal period [2]. Although CV port implantation is just a minor surgery and is not contraindicated under the product characteristics, bevacizumab should be withdrawn for a certain period. This is contraversial, but we have added this information to the introduction section (page 2, lines 68–72).
Reference:
[1]. Japanese Society of Interventional Radiology. The Official Journal of the Japanese Society of Interventional Radiology. 2021, 35, 359-397. DOI:10.11407/ivr.35.359
[2]. https://chugai-pharm.jp/content/dam/chugai/product/ava/div/guide-cl/doc/ava_guide_cl.pdf
- The rationale for dividing the two groups based on more or less than seven days between minor surgery and administration of bevacizumab should be explained. The summary of product characteristics mentions 28 days for major surgery.
Response: We thank the reviewer for this comment. Since it has been reported that early initiation of bevacizumab administration after CV port implantation increased the delay in wound healing, a guideline for proper use of bevacizumab in Japan recommend a one-week drug withdrawal period [1]. Whereas, previous report showed that there were no problems with the early administration of bevacizumab. In addition, guidelines for CV port implantation in Japan have not reached a consensus [2]. The withdrawal period has not been established yet; therefore, we aimed to verify the current one-week withdrawal period in this study. We have added this information to the Material and Methods section (page 2, lines 84–85).
Reference:
[1]. https://chugai-pharm.jp/content/dam/chugai/product/ava/div/guide-cl/doc/ava_guide_cl.pdf
[2]. Japanese Society of Interventional Radiology. The Official Journal of the Japanese Society of Interventional Radiology. 2021, 35, 359-397. DOI:10.11407/ivr.35.359
- Consider using time-dependent event analysis by constructing Kaplan-Meier curves for patients receiving regimens with and without bevacizumab and comparing them using logrank test. Instead of dichotomizing +/- 7 days, consider using time data as a continuous variable.
Response: According to the reviewer's comment, we have constructed Kaplan-Meier curves for patients receiving regimens with (n = 184) and without (n =57) bevacizumab and compared them using the log-rank test (Supplementary Figure S1).
- The study design being retrospective may limit the strength of the conclusions drawn.
Response: Thank you for this insightful comment. We have mentioned this information in the discussion section as a limitation (single-center, retrospective study) but to tone it down, we modified the conclusion (line 300).
Minor comments:
- The statement "BEV has shown antitumor effects in CRC and is one of the most important drugs in standard therapy for CRC" on lines 47-48 requires justification by referencing several phase III study data or modifications.
Response: Thank you for this insightful comment. We have added some appropriate references (line 51).
- Please verify the number and fraction of females on line 127.
Response: Thank you for this comment. I checked the number and stated the correct value.

Reviewer 2 Report
I have no further questions for the authors.
Author Response
Manuscript ID: cancers-2281323, “Safety of early bevacizumab administration after central venous port placement for patients with colorectal cancer”, by H. Shigyo et al.
We thank the reviewer for evaluating our manuscript.
Reviewer 3 Report
This manuscript is an original article that retrospectively investigated the safety of early bevacizumab administration after central venous (CV) port placement for patients with colorectal cancer. The authors showed that there were no significant differences in the frequency of complications between early and late administration group and concluded that early BEV administration after CV port placement was safe.
This study was conducted well, and the methods are appropriate. The data are presented clearly. In general, this is a well-written paper that presents interesting data. The results will be of interest to clinicians in the field.
However, the following minor issues require clarification:
Minor
1. The authors described that this study included patients who were pathologically diagnosed with unresectable CRC and who received chemotherapy with BEV for the first time after CV port placement. According to this definition, 57 patients whose chemotherapy regimen did not include BEV should be excluded from the first.
2. Please use an abbreviation of “GI” for “gastrointestinal”.
3. (P3L129-134) “(Table 1)” can be deleted.
4. (Figure 1) An arrow from “Chemotherapy was initiated within 7 days after CV port implant” can cause confusion. Please modify it.
5. The authors introduced two previous related study which results were conflicting [23-24]. I recommend that the authors describe the topic of this study is still unclear and controversial in the introduction section.
6. (Table 2, 3) Please emphasize statistically significant P-values to help readers’ understanding.
7. (Figure 2) This figure seems to provide only a little information.
Author Response
Manuscript ID: cancers-2281323, “Safety of early bevacizumab administration after central venous port placement for patients with colorectal cancer”, by H. Shigyo et al.
We thank the reviewer for evaluating our manuscript. The following text describes our responses to the comments.
Response to the comments of Reviewer 3
Minor
- The authors described that this study included patients who were pathologically diagnosed with unresectable CRC and who received chemotherapy with BEV for the first time after CV port placement. According to this definition, 57 patients whose chemotherapy regimen did not include BEV should be excluded from the first.
Response: Thank you for your insightful comment. We apologize for the confusion caused due to the discrepancies between the text and figure descriptions. To clarify, we revised the text in the Material and Methods section (page 2, lines 76–77).
- Please use an abbreviation of “GI” for “gastrointestinal”.
Response: According to the reviewer's comment, we used the abbreviation GI for 'gastrointestinal'.
- (P3L129-134) “(Table 1)” can be deleted.
Response: According to the reviewer's comment, we have deleted the “(Table 1).”
- (Figure 1) An arrow from “Chemotherapy was initiated within 7 days after CV port implant” can cause confusion. Please modify it.
Response: Thank you for this comment. To avoid confusing the readers, we modified the Figure (revised Figure 1).
- The authors introduced two previous related study which results were conflicting [23-24]. I recommend that the authors describe the topic of this study is still unclear and controversial in the introduction section.
Response: Thank you for your insightful comment. As per the reviewer's suggestion, we have added this information in the introduction section (page 2, lines 64–73).
- (Table 2, 3) Please emphasize statistically significant P-values to help readers’ understanding.
Response: Thank you for this comment. We emphasized (with bold font) statistically significant p-values to further readers’ understanding.
- (Figure 2) This figure seems to provide only a little information.
Response: Thank you for this insightful comment. To enrich the information, we have constructed Kaplan-Meier curves for patients receiving bevacizumab-containing regimens in the early/late administration groups and compared them using the log-rank test. We have added Figure 2B.

Round 2
Reviewer 1 Report
Thank you for submitting your revised manuscript and for addressing the major and minor comments in a sufficient way. Thanks for adding Kaplan-Meier curves.
The manuscript sheds light on a controversial issue and will be of interest to clinicians not only in Japan but also worldwide. The research and analysis have contributed to our understanding of this topic and may be helpful in clinical practice.